# OpenReview forum: "Decoding-Time Language Model Alignment with Multiple Objectives"
_NeurIPS.cc/2024/Conference — NeurIPS 2024 poster_

### Official Review · Reviewer_TTLg · 2024-06-26

**Soundness:** 3
**Presentation:** 4
**Contribution:** 3
**Rating:** 7
**Confidence:** 3

**Summary:**

The paper introduces a decoding-time alignment method called Multi-Objective Decoding (MOD) for aligning language models (LMs) with multiple objectives. MOD combines a set of models aligned for individual rewards and allows any weightings (even not-all-positive) for rewards. MOD leverages a common form among f-divergence regularized alignment approaches to derive an efficient decoding strategy that greedily selects the next token from an algebraic combination of predicted probabilities of all base models. MOD can maximize an interpolated reward function without extensive retraining. The paper theoretically demonstrates the sub-optimality of existing approaches and establishes optimality guarantees for MOD. Empirical results show MOD's effectiveness, with a 12.8% overall reward improvement over a parameter-merging baseline when optimizing for three objectives.

**Strengths:**

* **Flexibility in Objectives Alignment**: MOD's primary strength lies in its ability to align language models with multiple objectives simultaneously. This flexibility allows it to balance and prioritize different user needs and preferences without the need for retraining the model for each new objective combination.
* **Efficiency and Simplicity**: The algorithm is efficient and simple to implement, as it only need to operate the probabilities at decoding time.
* **Theoretical Robustness and Empirical Validation**: MOD is underpinned by a strong theoretical foundation, with proofs that demonstrate its optimality and sub-optimality of existing methods. The paper provides empirical evidence of MOD's effectiveness across various tasks and models, showcasing its practical applicability and robustness in real-world scenarios.

**Weaknesses:**

I don't see any major weaknesses in this paper.

**Questions:**

* Is the conditions of Eq. 6 correctly?
* Line 231-233: Can you explain more about this?

---

> ### Author Rebuttal · Authors · 2024-08-05
>
> **Q1**
>
> *Q: Is the conditions of Eq. 6 correctly?*
>
> A: We omit some details in the main content. The formal conditions of Eq. 6 are provided in **Appendix D.3 Theorem 5**.
> Briefly speaking, we require the $f$ to be strong-barrier function and the set of $\{\pi_{i=1,..,M}\}$ to be optimial for each reward (objective) function. The first condition can be always satisfied for all the existing regularization methods.
>
> For the second condition, we assume it is true to explain the main theoretical intuition. Later in **Appendix F**, we further show that our algorithm is robust even when those policies are suboptimal as long as they are not too bad.
>
> **Q2**
>
> *Q: Line 231-233: Can you explain more about this?*
>
> A: Yes. Since the optimization objective of general RLHF methods is $\mathbb E[\mathcal R]-\beta\text{KL}$, it is common to show $\text{KL}(\pi\vert\pi_{\text{ref}})$ in addition to rewards, to indicate that $\pi$ does not deviate much from $\pi_{\text{ref}}$. However, our approach through reformulation can only generate one response for the prompt, not obtaining an exact model. Therefore, it is not accessible to get the KL divergence value. As compensation, we show example generations to indicate that the obtained policy does not deviate much from the refence policy.

---

### Official Review · Reviewer_BvFb · 2024-07-12

**Soundness:** 3
**Presentation:** 3
**Contribution:** 3
**Rating:** 5
**Confidence:** 2

**Summary:**

The authors propose a decoding method that aims to combine the predictions of diverse models that are aligned with different objectives. In their multi-objective setting, the goal is to find an optimal policy that maximize a weighted, multi-objective reward, given the policy aligned to each of the individual rewards. In particular, the authors propose a reformulation using Legendre transform to bypass calculating Z (normalization) at a sequence level.

**Strengths:**

1. The authors provide detailed theoretical analysis to justify their approach.
2. The authors show that the proposed method can handle negative weights for rewards, which cannot be accomplished by previous work.

**Weaknesses:**

1. The baselines appears weak. For example, in Appendix F, the main comparison is against RS. However, RS cannot even outperform the best individual model in all experiments (Tables 7,8,9,10).
2. The proposed MOD also seems not much stronger. MOD can only beat the best individual model on 2/4 settings in Appendix F.
3. Lack of baselines. It would be helpful if the authors can include more generic ensemble baselines such as weighted averaging/voting.

**Questions:**

1. Has the authors studied the inference overhead of the proposed MOD?

**Limitations:**

Conclusion and throughout the work.

---

> ### Author Rebuttal · Authors · 2024-08-05
>
> **W1**
>
> *Q: The baselines appears weak. For example, in Appendix F, the main comparison is against RS. However, RS cannot even outperform the best individual model in all experiments (Tables 7,8,9,10).*
>
> A: We have pointed out the sub-optimality of rewarded soup in **Section 6.1**. Although it is a widely used retraining-free approach for multi-objective alignment, it can often lead to suboptimal performance. However, as shown in the original papers [1][2][3] and the experiments in **Section 5**, it is still a strong baseline.
>
> **W2**
>
> *Q: The proposed MOD also seems not much stronger. MOD can only beat the best individual model on 2/4 settings in Appendix F.*
>
> A: We would like to clarify that, the primary goal of **Appendix F**, is to examine the performance of MOD and RS when the base models are suboptimal (early-stopped). And the advantages of MOD over RS show the robustness of our method.
>
> Due to restricted computation resource, it is not easy for us to train a model with prominent performance on HelpSteer dataset (this is why we put it in appendix for reference purpose). And thus, certain base models would drag others down in the logit mixing process. For example, in **Table 3**, $\pi_{1f}$ is obviously trained overall better than $\pi_{2f}$, and $\pi_{3f}$, and thus combining them together would definitely be worse than $\pi_{1f}$ itself, but MOD is not much influenced, compared to RS, demonstrating its robustness.
>
> The performance of MOD with well-trained models is provided in **Section 5**.
>
> **W3**
>
> *Q: Lack of baselines. It would be helpful if the authors can include more generic ensemble baselines such as weighted averaging/voting.*
>
> A: The basic idea of voting is to find out the majority’s voted candidate, and thus requires the possible responses to be fewer than models. Therefore, it is often adopted in multiple choice questions, and not applicable in our tasks.
>
> We have compared MOD with several commonly used weighted averaging approaches, including *average ensemble*, and *perplexity-guided ensemble* (packLLM [4]). Common ensemble methods cannot flexibly generate customized responses, since their weighting selection rules are not related to preferences. Please see the **attached PDF** in global response for experimental results.
>
> **Q1**
>
> *Q: Has the authors studied the inference overhead of the proposed MOD?*
>
> A: Please see **global response**.
>
> [1] Rewarded soups: towards Pareto-optimal alignment by interpolating weights fine-tuned on diverse rewards. arXiv preprint arXiv:2306.04488.
>
> [2] Personalized Soups: Personalized Large Language Model Alignment via Post-hoc Parameter Merging. arXiv preprint arXiv:2310.11564.
>
> [3] Conditioned Language Policy: A General Framework for Steerable Multi-Objective Finetuning. arXiv preprint arXiv:2407.15762.
>
> [4] Pack of LLMs: Model Fusion at Test-Time via Perplexity Optimization. arXiv preprint arXiv:2404.11531.

---

### Official Review · Reviewer_pv2M · 2024-07-13

**Soundness:** 3
**Presentation:** 2
**Contribution:** 4
**Rating:** 8
**Confidence:** 4

**Summary:**

In many practical uses of RLHF the reward function is the convex combination of several rewards. Instead of training a single policy attempting to maximize the expected aggregate reward (subject to the usual regularization keeping it close to an anchor policy), the authors show that one can train separate policies, one for each reward and then mix them at decoding time using the same convex combination in log-probability space.

One important consequence is that one can change the weights on various rewards at decoding time, per response, making the algorithm very appealing for situations where the balance between certain rewards needs to change depending on the prompt/context.

**Strengths:**

Novel approach to dealing with rewards that are a linear mix of "elementary" rewards; mathematically sound.

Offers a simple, practical way of changing the mix of "elementary" rewards at decoding time, per model response.

**Weaknesses:**

The presentation could be much simpler, starting from the ubiquitous case of using KL divergence for regularization, which also leads to the elegant log-linear combination in Eq. (7). The general case for f-divergence could be mentioned, but relegated to the already prodigious appendix.

One technical weakness of the proposed approach is that one needs to serve/run M different policies at decoding time, which is significant overhead.

After completing the review I have become aware of the work in:
@misc{wang2024conditionedlanguagepolicygeneral,
      title={Conditioned Language Policy: A General Framework for Steerable Multi-Objective Finetuning},
      author={Kaiwen Wang and Rahul Kidambi and Ryan Sullivan and Alekh Agarwal and Christoph Dann and Andrea Michi and Marco Gelmi and Yunxuan Li and Raghav Gupta and Avinava Dubey and Alexandre Ramé and Johan Ferret and Geoffrey Cideron and Le Hou and Hongkun Yu and Amr Ahmed and Aranyak Mehta and Léonard Hussenot and Olivier Bachem and Edouard Leurent},
      year={2024},
      eprint={2407.15762},
      archivePrefix={arXiv},
      primaryClass={cs.LG},
      url={https://arxiv.org/abs/2407.15762},
}

Section 6 and Appendix E in [wang2024conditionedlanguagepolicygeneral] are directly relevant to this paper, deriving a sensitivity analysis for logit mixing, the log-linear combination in Eq. (7).

**Questions:**

The reformulation using Legendre transformation in Section 4.2 was hard to follow.

Again, a crisp derivation for the case of KL regularization that anyone can follow would greatly improve the reach, and implicitly impact of the paper.

**Limitations:**

One technical weakness of the proposed approach is that one needs to serve/run M different policies at decoding time, which is significant overhead.

---

> ### Author Rebuttal · Authors · 2024-08-05
>
> **W1**
>
> *Q: The presentation could be much simpler, starting from the ubiquitous case of using KL divergence for regularization, which also leads to the elegant log-linear combination in Eq. (7). The general case for f-divergence could be mentioned, but relegated to the already prodigious appendix.*
>
> A: Thank you for your suggestions! We would consider focusing on the reverse KL case in main content in later version.
>
> **W2**
>
> *Q: One technical weakness of the proposed approach is that one needs to serve/run M different policies at decoding time, which is significant overhead.*
>
> A: Please see **global response**.
>
> **W3**
>
> *Q: After completing the review I have become aware of the work in [1].*
>
> *Section 6 and Appendix E in [1] are directly relevant to this paper, deriving a sensitivity analysis for logit mixing, the log-linear combination in Eq. (7).*
>
> A: Yes, this paper is a great work and very relevant to our work. It is released on July 2024, while the submission date of NeurIPS is May 2024. Thus it is concurrent.
>
> Their sensitivity analysis on logit mixing is basically of the same idea as our **Section 6.3**. The main difference in implementation is that, they use parameter-merging as an approximation of logit mixing. Notably, we have pointed out the sub-optimality of parameter-merging in **Section 6.1**, but in [1], they begin to merge models during training, and thus alleviates this issue thanks to a good coverage of training data. Anyway, we are focusing on decoding-only generation tasks, so their solution involving training is not applicable to ours.
>
>
> **Q1**
>
> *Q: The reformulation using Legendre transformation in Section 4.2 was hard to follow. Again, a crisp derivation for the case of KL regularization that anyone can follow would greatly improve the reach, and implicitly impact of the paper.*
>
> A: More details of reformulation are provided in **Section D.3**. The key ideas are:
>
> 1) We don’t need to get an exact model to sample responses. Decoding one response for one prompt is enough.
>
> 2) The response selection can be viewed as a constrained optimization problem: to maximize reward, with a not too low probability for reference policy (or equivalently, to maximize probability for reference policy, with a not too low reward).
>
> 3) The reward can be represented using a mapping from predicted probabilities of base policies.
>
> 4) Finally we can get a closed form solution for the dual problem.
>
> And yes, we would consider showing a simple derivation of reformulation for the KL case in later version.
>
> **Limitations**
>
> *Q: One technical weakness of the proposed approach is that one needs to serve/run M different policies at decoding time, which is significant overhead.*
>
> A: Please see **global response**.
>
> [1] Conditioned Language Policy: A General Framework for Steerable Multi-Objective Finetuning. arXiv preprint arXiv:2407.15762.

---

### Official Review · Reviewer_dVwo · 2024-07-13

**Soundness:** 3
**Presentation:** 3
**Contribution:** 3
**Rating:** 7
**Confidence:** 3

**Summary:**

This paper presents Multi-Objective Decoding (MOD), a novel algorithm designed to align language models (LMs) with multiple human preferences simultaneously during decoding. MOD addresses the limitations of existing methods that optimize LMs for a single reward function, thereby providing flexibility and efficiency without the need for retraining. The authors define multi-objective reward functions and assume the existence of single-objective aligned LMs optimized for specific rewards. By leveraging the properties of strong-barrier functions and using the Legendre transform, they derive a closed-form solution for linearly combining the outputs of different models, achieving multi-objective alignment. This method guarantees optimality under certain conditions and transforms response-level decoding into efficient token-level decoding using greedy search. Extensive experiments validate MOD&#39;s effectiveness, demonstrating significant improvements in reward optimization compared to parameter-merging baselines.

**Strengths:**

MOD introduces a novel method for multi-objective alignment, enabling language models to align with multiple objectives simultaneously during decoding, thus eliminating the need for retraining.

The authors provide a robust theoretical framework by defining multi-objective reward functions and leveraging strong-barrier functions. They prove a closed-form bijection between single-objective models and their rewards, and derive a closed-form solution using the Legendre transform.

MOD achieves optimality guarantees under certain conditions and transforms response-level decoding into efficient token-level decoding using greedy search, making the method both effective and practical.

Extensive experiments demonstrate MOD's superior performance, showing a 12.8% overall reward improvement compared to parameter-merging baselines when optimizing for three objectives. The effectiveness is validated across various tasks and model sizes.

**Weaknesses:**

Although MOD circumvents the need for retraining, it requires loading multiple models concurrently, which can be computationally intensive and may not scale efficiently for a larger number of objectives or bigger model sizes.

The paper could benefit from a more detailed discussion on potential negative impacts or failure modes, especially in scenarios involving conflicting objectives or suboptimal base model alignment.

**Questions:**

How does the MOD approach scale with an increasing number of objectives? Are there any practical limits to the number of objectives that can be managed simultaneously?

Can the authors provide more insights into the sensitivity of MOD to the quality of base models? Specifically, how does the performance degrade if the base models are not well-aligned or are suboptimal?

Are there any guidelines or best practices for setting and adjusting these preference weightings to achieve optimal results?

**Limitations:**

The authors have addressed the limitations in Section 7.

---

> ### Author Rebuttal · Authors · 2024-08-05
>
> **W1**
>
> *Q: Although MOD circumvents the need for retraining, it requires loading multiple models concurrently, which can be computationally intensive and may not scale efficiently for a larger number of objectives or bigger model sizes.*
>
> A: Please see **global response**.
>
> **W2**
>
> *Q: The paper could benefit from a more detailed discussion on potential negative impacts or failure modes, especially in scenarios involving conflicting objectives or suboptimal base model alignment.*
>
> A:
> - Yes, we have provided sensitivity analysis (scenarios involving suboptimal base models) in **Section 6.3**.
> - As for conflicting objectives, we have shown in **Section 4.2** and **Appendix D.3**, that MOD can generate a response with the interpolated reward $r\ge C_2$ (where $C_2$ is a threshold based on $\pi_{\text{ref}}$ and the reward distribution), and this is not affected by the relationship between these objectives. For example, let us take $\mathcal R_1$:Helpful and $\mathcal R_2$:Harmless. Then a potential issue might be, there may not exist a response that is both very helpful and very safe. But our approach can guarantee that we obtain a response with $w_1\mathcal R_1+w_2\mathcal R_2$ larger than that threshold. We will include this explanation in a later version.
>
> **Q1**
>
> *Q: How does the MOD approach scale with an increasing number of objectives? Are there any practical limits to the number of objectives that can be managed simultaneously?*
>
> A: Theoretically, there is no limits in the number of objectives, while the time/space cost will increase. Please see **global response** for details of how to alleviate this issue. In practice, the number of objectives cannot be too large [1][2][3], and is usually restricted to fewer than 5. Therefore, the additional inference cost is still acceptable for the provider of LLMs, who is capable of supporting large-scale MoE systems [4][5].
>
> **Q2**
>
> *Q: Can the authors provide more insights into the sensitivity of MOD to the quality of base models? Specifically, how does the performance degrade if the base models are not well-aligned or are suboptimal?*
>
> A: Please see W2.
>
> **Q3**
>
> *Q: Are there any guidelines or best practices for setting and adjusting these preference weightings to achieve optimal results?*
>
> A: There can be two definitions for optimal results based on two different settings.
>
> 1) Customized response generation based on user preference. In this setting, there is no gaurantee that we can get improvement on every objective. Instead, we care about finding a model that matches the user preference. In **Section D.3**, we have gaurantees that, weightings $w_1,w_2,\ldots,w_m$ during policy decoding time are the best combination for reward function $\sum_{i=1}^m w_i\mathcal R_i$. In the other word, as long as the weighting chosen by the user matches their true performance, we can gaurantee a model that matches the users perference. And intuitively, if all the model are trained under the *same reward scales*, then the weights chosen by users should righlty reflect their preferences on these objectives. As demonstrated in **Section 5.2 Figure 2,3,4**, we believe this "same scale" assumption is effective in practical.
>
> 2) Faster model combination for general performance improvements. In this setting, we aim to find a generally better model via adjust the weights in decoding time only. As shown in **Section 5.2 open-instruction following**, there may not be a fixed optimal solution, but we can *quickly* experiment with some weightings, and then test on certain benchmarks. Since our method is greedy and decoding-time-only, this pipeline is very efficient.
>
> [1] UltraFeedback: Boosting Language Models with Scaled AI Feedback. arXiv preprint arXiv:2310.01377.
>
> [2] HelpSteer: Multi-attribute Helpfulness Dataset for SteerLM. arXiv preprint arXiv:2311.09528.
>
> [3] Beyond One-Preference-Fits-All Alignment: Multi-Objective Direct Preference Optimization. arXiv preprint arXiv:2310.03708.
>
> [4] Language Models are Few-Shot Learners. arXiv preprint arXiv:2005.14165.
>
> [5] A Closer Look into Mixture-of-Experts in Large Language Models. arXiv preprint arXiv:2406.18219.

---

### Author Rebuttal · Authors · 2024-08-05

We appreciate all the reviewers for their timely and positive feedbacks. Reviewer dVwo describes our approach, MOD, as “novel, effective and practical”, and notes that “the theoretical framework is robust”; Reviewer pv2M thinks MOD is “novel, simple, practical, and mathematically sound”; Reviewer BvFb acknowledges our contributions to “detailed theoretical analysis” and praises MOD’s excellence in handling negative weights; Reviewer TTLg highlights the “flexibility, efficiency and simplicity” of MOD, acknowledging the “theoretical robustness and empirical soundness”.

Reviewer TTLg thinks there is no major weakness in this paper. Other three reviewers (dVwo, pv2M, BvFb) raise concerns about the inference overhead of MOD as the increase of the number of objectives. We would like to clarify this point in detail: indeed, there is a trade-off between performance and time/space cost, similar to the deployment of MoE systems [1][2]. And there are two ways alleviating this issue (visualizations are provided in the **attached PDF**):

1) Using light-weight adapters. Let $m$ be the number of objectives. Then by training one adapter for one objective, we can implement MOD with $\mathcal O(1)$ space and $\mathcal O(m)$ time cost in inference time, compared to one model. For example, a recent manuscript [3] adopts the same idea in reward model training.

2) Distributed deployment. Note that we can first compute the logits of each model at the same time, which is parallelable, and then do the mixing, thus MOD can be implemented with $\mathcal O(m)$ space and $\mathcal O(1)$ time cost in inference time. Furthermore, we expect that this can be further accelerated with the development of large-scale ML-systems like [4][5].

We are happy to address any further comments from reviewers.

[1] Language Models are Few-Shot Learners. arXiv preprint arXiv:2005.14165.

[2] A Closer Look into Mixture-of-Experts in Large Language Models. arXiv preprint arXiv:2406.18219.

[3] Interpretable Preferences via Multi-Objective Reward Modeling and Mixture-of-Experts. arXiv preprint arXiv:2406.12845.

[4] DeepSpeed4Science Initiative: Enabling Large-Scale Scientific Discovery through Sophisticated AI System Technologies. arXiv preprint arXiv:2310.04610.

[5] Efficient Memory Management for Large Language Model Serving with PagedAttention. arXiv preprint arXiv:2309.06180.

---

### Decision · Program_Chairs · 2024-09-25

**Decision:**

Accept (poster)

**Comment:**

This paper proposes a novel multi-objective decoding (MOD) technique, it assumes that the reward function of RLHF is a convex combination of multiple reward functions, establishes theoretical optimality guarantees, and eliminates the need for extensive retraining. This paper is solid in its presentation, theory, and experiments, and has received positive comments from all reviewers. Regarding the drawbacks, the main issues are computational overhead and scalability. Overall, this paper deserves to be accepted.